# “Weaving a Mat That We Can All Sit On”: Qualitative Research Approaches for Productive Dialogue in the Intercultural Space

**DOI:** 10.3390/ijerph19063654

**Published:** 2022-03-19

**Authors:** Emma Haynes, Minitja Marawili, Alice Mitchell, Roz Walker, Judith Katzenellenbogen, Dawn Bessarab

**Affiliations:** 1School of Global and Population Health, University of Western Australia, Crawley, WA 6009, Australia; roz.walker@uwa.edu.au (R.W.); judith.katzenellenbogen@uwa.edu.au (J.K.); 2Centre for Aboriginal Medical and Dental Health, University of Western Australia, Crawley, WA 6009, Australia; dawn.bessarab@uwa.edu.au; 3Menzies School of Health Research, Casuarina, Darwin, NT 0810, Australia; minitja.marawili@gmail.com (M.M.); alicem404@gmail.com (A.M.)

**Keywords:** health inequalities, co-design and community engagement, First Nations, Australian Aboriginal, Socially Disadvantaged Communities, innovative research practices, decolonising methodologies, intercultural, productive dialogue

## Abstract

Research remains a site of struggle for First Nations peoples globally. Biomedical research often reinforces existing power structures, perpetuating ongoing colonisation by dominating research priorities, resource allocation, policies, and services. Addressing systemic health inequities requires decolonising methodologies to facilitate new understandings and approaches. These methodologies promote a creative tension and productive intercultural dialogue between First Nations and Western epistemologies. Concurrently, the potential of critical theory, social science, and community participatory action research approaches to effectively prioritise First Nations peoples’ lived experience within the biomedical worldview is increasingly recognised. This article describes learnings regarding research methods that enable a better understanding of the lived experience of rheumatic heart disease—an intractable, potent marker of health inequity for First Nations Australians, requiring long-term engagement in the troubled intersection between Indigenist and biomedical worldviews. Working with *Yolŋu* (Aboriginal) co-researchers from remote Northern Territory (Australia), the concept of *ganma* (turbulent co-mingling of salt and fresh water) was foundational for understanding and applying relationality (*gurrutu*), deep listening (*nhina, nhäma ga ŋäma),* and the use of metaphors—approaches that strengthen productive dialogue, described by *Yolŋu* co-researchers as weaving a ‘mat we can all sit on’. The research results are reported in a subsequent article.

## 1. Introduction

Research remains a site of struggle over knowledge claims and power for First Nations (see Box 1) people globally. Critical Indigenist theorists argue that the concepts and practices of Western research traditions are deeply bound up with histories of colonisation [1]. Certainly, there is little to recommend research approaches that privilege dominant medical narratives and interactions as a superior and more valid form of knowledge reflected in theories of knowledge, and highly specialised forms of language [2].

Box 1Terminology.Terminology regarding identity is varied and complex. We use the term ‘First Na-tions’ where we discuss international concepts and reflecting an international readership (except where we retain the terms preferred by cited authors). Aborig-inal and Torres Strait islander peoples are the ‘First Nations’ of Australia. In this paper, we use the term Aboriginal when describing the communities, co-researchers, and participants in this study. No disrespect is intended to Torres Strait Islander peoples or the diverse cultures, histories, and languages of other First Nations Australians. Yolŋu-speaking Aboriginal co-researchers used the term Yolŋu to refer to all Aboriginal people, and, where appropriate, we have retained this. Balanda is the Yolŋu term for non-Aboriginal people. 

Such traditional positivistic or medicalised approaches reinforce existing dominant structures of power and control, thereby perpetuating the ongoing impacts of colonisation, shaping research priorities, resources allocation, policies, and models of care [3,4]. Critically, this results in a form of social control, without the need for overt coercion, through a tendency to blame First Nations peoples for their poorer health and an assumption that health problems can only be fixed by ‘the ingenuity, expertise and generosity of the outsider’ [5]. Such approaches are privileged at the expense of examining health system, socio-economic, historical, and political factors [6]. 

Concurrently, despite the depth, longevity [4], and legitimacy of First Nations peoples’ research knowledges and methodologies [7], sustained efforts to challenge Western hegemony and demand sovereignty in the research space continue to be constrained. Consequently, Western researchers have largely retained control over the initiation, process, evaluation, and dissemination of research [8,9,10]. The failure of researchers and policy makers to meaningfully address inequities or acknowledge First Nations peoples’ ways of knowing, being, and doing [11] can result in ‘collateral damage’ such as “stigma, internalised blame, emotional suffering, and hypervigilance that reproduces structural violence” [12].

### 1.1. New Approaches Are Needed

Addressing the systemic injustices that underpin inequities in First Nations peoples’ health requires new knowledges and understandings based on decolonising methodologies and a critical understanding of power dynamics. Decolonising methodologies that de-centre Western epistemologies seek to privilege the concerns, practices, and participation of First Nations people as researchers [13]. Rather than continuing the ‘politics of polarity’ [14], First Nations authors seeking to constructively address the impacts of colonisation have variously described an intersection between First Nations and mainstream worldviews, using terms such as the cultural interface [15], liminal, intercultural space [16], third space [17,18], hybrid space, and borderlands. Regardless of terminology, this space is envisaged as enabling new positions to emerge by engaging in discussion and negotiation to explore new ideas and ways of working, facilitating the creation and emergence of new knowledge and understandings [19,20]. Where one system is not superior over the other [21] innovation can occur to fundamentally improve understanding in both knowledge systems [4] as researchers collaboratively weave back and forth between their worldviews. As binaries are deconstructed, it becomes a space ‘where Western and Aboriginal converge and constitute each other’ [22] in a process of productive dialogue. 

More than just a ‘middle ground’, the idea of productive dialogue can be traced to ancient Greek, Native American, and other Indigenous cultures, as a unique kind of conversation that features people listening deeply to each other, ‘deepening connectedness, building trust, and a willingness to disclose’ [23]. It also reflects a broader poststructuralist approach that seeks to shift ‘either/or’ opinions to ‘both/and’ dialogue [24]. As productive dialogue assumes engagement in rigorous discussion to share ideas, experiences, and understandings, it can result in a struggle between two knowledge systems, exposing tensions that can be risky and require attention to power differences [19]. Productive dialogue can only happen when any potential power differences are recognised and acknowledged and all participants feel equally included, valued, and heard. The space of productive dialogue can and should be a place of creative tension and this requires doing ‘the necessary work of choosing’ to put oneself in what has been termed ‘the discomfort zone’ of intercultural work [25]. It is only when the productive potential of difference is emphasised that the actual work of collaboration is achieved [24].

In parallel with developments at the cultural interface, the potential of sociology, medical anthropology, and community participatory action research (CPAR) approaches to effectively mitigate the impacts of colonisation through better integrating patient lived experience into the biomedical worldview is increasingly recognised [7]. Such developments are both critical and urgent in order to interrogate and reveal associations between colonisation, racism, poverty, and poor health. Failure to conduct research aimed at unravelling the mechanisms of poverty leaves “the root causes unchanged” ([26], p. 11). Finally, while narrative and stories are important sources of data and transformative experiential knowledge, critical theory ultimately ‘holds an explanatory power’ ([27], p. 441). Critical theorists attend to power dynamics, creating ‘space for formerly marginalised peoples to step inside’ to engage and contribute to the dialogue about their own ways of knowing, being and doing ([13], pp. 165–169). 

### 1.2. Rheumatic Heart Disease Research as a Case Study

Rheumatic heart disease (RHD) is a potent marker of health inequity for Aboriginal and Torres Strait Islander Australians (hereafter Aboriginal). Despite the extensive medical, research, and policy initiatives to address the high rates of RHD experienced by Aboriginal Australians compared with non-Aboriginal Australians, there has been minimal sustained reduction [28,29]. The complex disease trajectory, lengthy secondary prevention, and surgical treatment for people living with RHD requires a long term and unavoidable engagement with the troubled cultural intersection between Aboriginal and Western/biomedical worldviews about health and wellbeing [30,31]. The commitment and advocacy of researchers and health practitioners will continue to be constrained by systemic and structural impediments while RHD remains increasingly understood in biomedical and epidemiological terms and where Aboriginal knowledges, lived experiences, and opinions have only recently become a priority in RHD research [32]. 

Findings from projects interviewing RHD key stakeholders [10,31] identified the dominance of a biomedical worldview in the RHD space and highlighted systemic power inequities that constrain the space of productive dialogue. Similarly, the findings of a systematic review of the literature related to the lived experience of RHD [32] provided evidence of the impact of a narrow biomedical research focus as only eight of the 15 Australian publications reviewed included qualitative research and only three had an exclusive focus on and specific interest in the Aboriginal voice. Most of the reviewed publications were published in biomedical journals, had authors trained in biomedicine, and were focused on solving biomedical issues—in particular, compliance with long-term injections. Researchers were primarily driven by pragmatic goals, addressing functional, managerial, or evaluation questions, such as how to ensure patients stayed on target with long-term secondary prophylaxis by assessing barriers to and enablers of care, evaluating interventions aiming to better manage people with RHD or measuring service planning, program development, and costs [32]. 

The publications that sought to identify barriers to adherence expressed a commonly reinforced view that noncompliance was a failure of families to provide care, with perceived parental confusion attributed to a direct shortcoming of responsibility [33]. Such research tends to lay the “blame for non-compliance on the Indigenous clients, establishing a cycle of suspicion” ([33], p. 204), rather than identifying how the health system fails to provide accessible, acceptable, and/or effective services for Indigenous clients, or focusing on features of the environmental, political, or economic systems that produce ill health and inequity [34]. 

In this way RHD research provides a powerful example where ‘the language of non-compliance is bound up with medical authority and control’ ([32], p. 16), contributing to asymmetric power relationships between staff and patient, exacerbated by health care providers who represent another culture and political power [10,30]. Such an approach fails to see actions, such as self-discharging from hospital, as an assertive act, a form of ‘micro-resistance’ [35] to hegemonic medical power [1,36], or acknowledge that the enduring effects of colonisation may explain “beliefs or behaviors such as mistrust in medical advice” [33,37].

### 1.3. Yolŋu Context and Concepts Related to Productive Dialogue in the Intercultural Space 

The research described here took place with the Yolŋu people in the north-eastern tip of Arnhem Land, Northern Territory. Settled in 1972, Yilpara is the largest homeland community in the region, but still a very small community of approximately 100 people (see Figure 1). As part of the ‘homelands movement’, the history of the Yilpara community is strongly linked to Aboriginal self-determination. The movement was an Aboriginal-initiated action demonstrating ‘the desire of Aboriginal people to assert control over their lives’ ([38], p. 343). The movement was also supported in that era by federal government self-determination policies, following the first land rights claims in the mid-1960s. Features of homeland communities include a desire to uphold traditional law and culture, a desire to escape from the dysfunction associated with living in more mainstream environments, and a rejection of government assimilationist policies. Through this movement Aboriginal leaders ‘sought to reassert the management and ownership of their land’ ([39], p. 114). The beneficial impact of homelands movement on health outcomes has been demonstrated, with lower-than-expected morbidity and mortality in a decentralised Aboriginal community [40].

Yilpara was featured as a case study of self-determination by anthropologists Frances Morphy and Howard Morphy [41]. Frances Morphy describes how both Yolŋu and mainstream systems of governance have relatively similar degrees of autonomy. Therefore, while the Yolŋu are encapsulated by mainstream systems, their autonomous actions produce effects on the system that can neither be categorised as ‘adaptation’ or ‘resistance’ and ‘is heavily imbued with Yolŋu principles of governance’ ([42], p. 132). Thus, both systems remain relatively mysterious to each other, and dialogue is constrained on both sides by a multitude of factors, the foremost being language and cultural worldview. The Commonwealth Government’s 2007 ‘emergency intervention’ in the Northern Territory with far-ranging impacts was an extraordinary social policy event that reinforced the subjugated social position of the Aboriginal people in the NT [30]. It reflected a tendency by policy makers to co-opt concepts such as the ‘intercultural space’ to imply a ‘space where Aboriginal people gradually merge with the mainstream’ ([43], p. 177). In this policy discourse, the expertise and viewpoints of the non-Aboriginal side of the intercultural context/interface becomes in effect the default, while the Aboriginal side is framed in deficit terms, thereby positioning Aboriginal people in a liminal or implied transitioning space, lacking agency and self-determination. 

In an early example of applying a traditional and profound First Nations’ concept in the intercultural space, the Yolŋu concept of *ganma* was shared by Yolngu leaders in the early 1990s to provide a place-based model of productive dialogue [44,45,46]. *Ganma* describes the mixing of salt water from the sea and fresh creek waters; it not only refers to brackish water but also to where two or more ideas, stories, or peoples are intermingled in an unconditional way. Developing and implementing *ganma* as a research process was described early on as an act of self-determination, just as the *Balanda* research has always taken a side but never revealed this, always claiming to be neutral and objective. *My aim in ganma is to help, to change, to shift the balance of power* ([44], p. 138; [47], p. 103 emphasis added). 

As a productive dialogue metaphor *ganma* fundamentally depends on Yolŋu and *Balanda* ‘identifying, respecting and maintaining differences, working collaboratively, coming to agreement, and building agreed ways of knowing and going ahead together’ ([48], p. 257; [49]). That is, the Yolŋu do not regard the *ganma* comingling of the waters as creating a solution, or something that weakens the two waters. On the contrary, they regard the co-mingling as similar to an emulsion, where both elements are intensely present—*ganma* is a place where each element is at its most intense, since each is most itself when it is in the presence of the other ([45], p. 138). *Ganma* has been identified in a Western research context as similar to Freire’s critical transformational framework based on dialogical knowledge co-creation processes [50,51]. Yolŋu co-researchers use the term *ganma* interchangeably with ‘both-way’ learning as more routinely used in conversations with *Balanda* (The currency of the term ‘both-way learning’ was expressed in https://www.rhdaustralia.org.au/news/graduating-country-both-way-academia (accessed on 31 January 2022). For the purposes of this article, productive dialogue, ganma and both-way learning are considered synonymous.). 

We report here on experiential learnings from developing and implementing research approaches and tools (methodologies and methods) in a project that sought to better understand the lived experience of RHD. The research was undertaken in collaboration with Yolŋu co-researchers from remote Northern Territory. Through a commitment to both-way learning, we created a space of productive dialogue together, described by Yolŋu co-researcher Makungun Marika as weaving a ‘mat we can all sit on’. A companion article published in this edition of the journal provides the research findings that resulted from applying these learnings [52]. 

## 2. Materials and Methods

### 2.1. Decolonising Qualitative Research

While attempting to address the limitations of traditional positivistic and medical approaches non-Indigenous qualitative researchers are equally at risk of being constrained by disciplinary knowledges and shaped by ‘imperial legacies; ongoing settler colonial relationships; unequal class, race, and gender divisions; and an increasingly corporatised university culture’ ([53], p. 316). Qualitative researchers are critiqued for a tendency to appropriate the ‘voice’ of their participants, the question of who benefits from research and the colonising potential of anthropology [53]. 

Acknowledging these critiques, the intention to work in the intercultural space on the topic of RHD was combined with a desire to extend standardised Western qualitative research methods to reflect a decolonising stance and privilege Aboriginal ways of knowing being and doing within the research activities. Adopting a decolonising stance, we deliberately employed two strategies to widen the space of both-way learning—reflexivity (including critical thinking about power dynamics) and building relationships and trust with Yolŋu co-researchers.

### 2.2. Reflexivity 

Reflexivity requires researchers to reflect critically on how their worldview (sociocultural positions, assumptions, and knowledge) influences the broad range of perspectives, activities, and outcomes that make up research processes. This is critical in establishing the rigor and trustworthiness of qualitative research [54]. Throughout the project described here, Aboriginal and non-Aboriginal academic researchers worked collaboratively and iteratively to achieve this. The non-Aboriginal researchers acknowledge the critical input of the senior academic Aboriginal researchers in facilitating reflexivity and engagement in ongoing courageous conversations [55] throughout the research process. This involved consciously engaging in critical reflection regarding a system that privileges both biomedical and non-Aboriginal worldviews.

### 2.3. Yolŋu Co-Researchers

Prior to commencing the project described here, building on existing relationships initially formed by co-author AM, the first author (EH) was invited to live in Yilpara for 12 months to co-ordinate a CPAR project [50]. Concurrently, co-authors MM and YG were participants in a Cert II in community health research associated with that project. These were significant steps in the process of moving beyond Western research methodologies as CPAR approaches encourage the use of locally-derived (in this case Aboriginal) research methodologies. Learning how to be in Yilpara (see findings below regarding relationships) led to further learnings about how to proceed with research in this context. That is, the commitment to first engage with and learn from the Yolŋu in a broad community context resulted in an immersion in Yolŋu ways of knowing, being, and doing, where relationships are central. From these relationships, trust was built, and Yolŋu co- researchers were later engaged in the lived experience of RHD research data collection, analysis, and dissemination of outcomes, ensuring that Yolŋu knowledges were privileged throughout the research. The Yolŋu co-researchers were a bridge to many data collection opportunities (observations and interviews), and their contribution to coding the yarns and identifying the themes in an iterative thematic analysis process was integral in shaping research findings. For non-Aboriginal co-researchers, such processes require lengthy community immersion and a commitment to building Yolŋu research capacity and confidence. The details of the research sample, data collection, and analysis methods are given in detail in our companion article [52]. 

### 2.4. Ethics

Ethics approval was obtained from the Human Research Ethics Committee of the NT Department of Health and Menzies School of Health Research. Approval number HREC 2016-2678. Reciprocal approval by UWA Human Research Ethics Office, Ref: ROAP 2020/ET000283. The following approvals were obtained prior to commencing the research: Ochre card (NT Working with Children); Northern Land Council (permit to reside in a homeland); and written approval from the Aboriginal elders of the community and the local Health Ser-vice.

## 3. Results

We report here on three Yolŋu concepts that can be applied as research approaches (both methodologies and methods): Relationality; *nhina, nhäma ga ŋäma* (sit with, listen, and observe); and metaphors as basis for critical and conceptual thinking. As with *ganma*, these concepts were widely used and accepted traditional Yolŋu ways of knowing, being, and doing, existing prior to any discussion of research. Through critical reflection and immersive, experiential learning, we identified how Western research methods could be aligned to Yolŋu concepts and applied in a research context. That is, as Marriot et al. [56] recommend, while keeping First Nations ways of knowing central, they also can be productively connected with Western epistemologies. Thus, each of the three Yolŋu concepts is described first in a broader more theoretical context and positioning in terms of First Nations research approaches, and then examples are given that demonstrate how Western research methods might be considered in alignment. As decolonising researchers, we recognise that First Nations assumptions and perspectives and mainstream research methodologies are not mutually exclusive and can be used effectively to complement and support each other ([57], p. 2). For the future, we see the learnings reported here as guiding a process of acknowledging and merging different knowledge systems through well-articulated and respectful consideration ([56], p. 5).

### 3.1. Relationality and Responsibility (Gurrutu)

Aileen Moreton-Robinson, in her review of the First Nations social research methods of Canada, the United States, Hawaii, Australia, and New Zealand, concluded that in different but similar ways Aboriginal ‘knowledge systems are grounded in relations to land, place, entities, ancestors, creators and people’. From this foundation she identifies the concept of ‘relationality’ as the ‘interpretive and epistemic scaffolding shaping and supporting Aboriginal social research’ ([16], p. 71). The concept of relationality is in contradistinction to the Western logic of discovery, which privileges a disconnected, human-centred, detached researcher ‘observing from a neutral position … [that] also requires being disconnected from the living earth’ ([16], p. 71). 

Morton-Robinson identifies a variety of relationships to be considered, such as those between the Aboriginal researcher and the Aboriginal community; the Aboriginal community and the researcher; the Aboriginal researcher and the Aboriginal academic community; non-Aboriginal researchers and the Aboriginal community; and between the academic community and Aboriginal methodologies ([16], p. 74). These relationships include conversation, support, guidance, and collegiality and contain specific responsibilities that guide knowledge and action and include a genuine desire to serve others. Responsibility is inseparable from respect and reciprocity ([16], p. 74).

Identity for the Yolŋu centres on their place in a circle of family relationships, a system of skin, clan, country, and family relationships termed *gurrutu*. Through their *gurrutu* connections and their local residence, Yolŋu people ‘know the country and are known by it, they speak the language and their sweat is familiar’ ([58], p. 168). This results in the very different social reality of Yolŋu wherein each person is


*‘understood not only as an individual personality but also having obligations and rights on a matrix of kinship and clan relations that constantly and concretely signal the connection and locatedness of everyone in the highly differentiated social whole’.*

*([44] , p. 46)*


Working out how you fit into this matrix is essential; when it is not clear how someone is related to you, much time is spent figuring it out and a ‘placeholder’ relationship might be assigned based on your totem, *mälk* (skin group), or place you are from (all of which have associated *gurrutu* connections). This is an ahistorical view—ancestors are contemporaneously present, felt, and taken into account. 

*Gurrutu* dictates that there are both people and places that are right to sit with (on) and those that are not. *Gurrutu* also determines the dances you can dance and the designs you can paint (what knowledges you are authorised to know) [59,60]. *Gurrutu* is foundational to learning, determining what knowledges can be taught, by whom and to whom, and when and in what formats (painting, dance, or music, song). 

For an outsider, the iterative process of learning about *gurrutu* allows an ever-increasing circle of connections to be established. That is, connections within a small family group can quickly be extended to others. Relationships with Yolŋu community researchers was the core of the lived experience of RHD research as they guided learning about Yolŋu ways of knowing, being, and doing. A strong foundation in *gurrutu* enabled the development of relationships with the community members who became thesis co-researchers, and the broader Yilpara community who acted as a *de facto* reference group for the research. Trusting relationships meant being confident to share significant life world experiences—for example, weeks of *bäpurru* (funeral) and *dhapi* (initiation) ceremonies and *Balanda* conferences were equally challenging both-way learning experiences and highly emotional times of bonding and learning. Significantly, in a concurrent project related to developing community research skills, as *gurrutu* informs correct behaviour it was therefore quickly equated by the Yolŋu with research ethics. 

### 3.2. Alignment with Non-Aboriginal Research Methods 

The capacity for empathy is often seen as an essential characteristic of a good qualitative researcher, related to both the quality of the research and the motivations for undertaking particular research topics. Thus, relationality is discussed here in terms of alignment with aspects of interviewing (yarning), thematic analysis and CPAR.

The concept of ‘yarning’ is a common feature of oral traditions for knowledge sharing, a form of conversation which utilises storying to impart information. Yarning, in this context, does not mean ‘just chatting about nothing of much importance’, as is implied in general Western uses of the term. It is a recognised technique used by Aboriginal Australians to connect, make meaning, and pass on knowledge and history, socially or more formally [61]. Yarning engages with Aboriginal processes such as respect, relationality, sharing, listening, and, in some cases, such as clinical yarning, the use of metaphors to explain and describe complex medical conditions. Yarning has been theorised and applied in a research context as 


*an informal and relaxed discussion through which both the researcher and participant journey together visiting places and topics of interest relevant to the research study. Yarning is a process that requires the researcher to develop and build a relationship that is accountable to Aboriginal people participating in the research … To have a yarn is not a one-way process but a dialogical process that is reciprocal and mutual.*

*([61] , p. 38)*


The strength of yarning is its capacity for relational meaning-making, thus broadening the space of productive dialogue. The key elements of dialogue, storytelling, and use of metaphor allow the negotiation of identities, roles, and relationships, rather than just conveying information—that is, yarning encourages empathy ([62], p. 85). Yarning requires deep listening skills.

Alongside participant observation (see next section) the lived experience of RHD research was based on yarns, many that were lengthy, detailed, and from the heart, involving participants who strongly felt they had a story to tell and were appreciative of the opportunity to share their experience to help others. In common with participation in CPAR, where the research process is collaborative and negotiated with participants [63], relationships are a source of motivation for participating in yarning research. The willingness to be a research participant on the basis of helping others was also reflected in the commitment of the Yolŋu researchers to the research process based on their relationships, in particular with others who had RHD experiences. For example, co-author MM has close family members with RHD, including her son and, significantly, a woman who was her mother-by-*gurrutu* who passed away from heart failure complications during the research period. It was these relationships, as well as their love for the community more broadly, that were frequently emphasised by the Yolŋu as the reason for working on the project. 

Later, as we read through or listened to the yarns (which the co-researchers had sometimes been involved in conducting) and discussed what they thought were the important ideas, the value of their ‘insider’ knowledge quickly became evident. This was not just cultural knowledge but also their own personal or family experiences of RHD. As a consequence of this understanding of, and immersion in, the lived experience of RHD and a world view that prioritises family/community, the Yolŋu were quick to arrive at themes, the connecting narrative, and also identifying conclusions and recommendations. This rapidity can potentially challenge a non-Aboriginal research expectation of ‘rigour’ that has a tendency to a pedantic and positivist scientific mind set about getting every detail and nuance of words coded, rather than accepting the immediate bigger picture thinking of the Yolŋu. 

### 3.3. Nhina, Nhäma ga Ŋäma (Sit with, Listen and Observe) 

A traditional Yolŋu term, ‘*nhina, nhäma ga ŋäma’* is literally translated as meaning to sit with, observe and listen (See [64] for a detailed discussion of each the components of *nhina, nhäma ga ŋäma* (sit, observe, and listen) to get a sense of the depth of meaning inherent in this phrase.). The broader concept refers to a way of learning, a way of being, and way of healing. It can be passive or active—a meditation, or a basis for dialogue or relationship. *Nhina, nhäma ga ŋäma* suggests a full understanding that occurs quietly and respectfully when one is in the right place, watching, and listening. It also implies waiting for a story to come to the hearer—we listen and observe and, if we are lucky, we learn something. For a non-Aboriginal researcher, it provides an essential guide for a way we can come to understand ourselves, others, and our world through feelings. Each component term holds great depth of meaning. Combined, *Nhäma* and *Ŋäma* ‘express extraordinary empathy, to the point that one can almost “become” the sufferer and feel what they are feeling “from within”’ ([65], p. 246); it is an ethical kind of relatedness that has ‘to do with mutual feelings of accountability and obligation, care, compassion and intimacy’ ([65], p. 256). It is for these reasons that Yolŋu co-researcher MM said ‘If we didn’t do this we couldn’t have done the rest, it’s the right way to work’ (Researcher journal note, November 2017). 

Until now, *Nhina, nhäma ga ŋäma* has not been documented as an Aboriginal research method. Building on the concept of *dadirri*, initially articulated in English by Miriam Rose Ungunmerr-Baumann (2002), a respected elder and artist from the Daly River region of the NT, it is increasingly referenced as a method of deep listening for action [2,56,66]. Taking *dadirri* as a starting point ‘expands the researcher’s worldview and opens our ears to other Aboriginal epistemologies and how we might engage with them in respectful and ethical ways’ ([66], p. 228). However, *nhina, nhäma ga ŋäma* adds new layers to the approach by including the significance of the land and people one sits with and the critical elements of relationship (*gurrutu*), empathy and observation (also related to reflexivity). 

As a form of learning, the concept of *nhina, nhäma ga ŋäma* is illustrated in the telling of stories to pass on knowledge in a way that allows the hearer to draw their own conclusions. For example, traditional stories, often referred to as dreaming stories are frequently told in different ways and contexts with different parts of the story told relevant to the ‘lesson’. Other than this selective telling, it is up to the hearer to understand the intended learning, encouraging reflection. In this context, the hearer is not expected or encouraged to ask questions, as Tyson Yunkaporta describes: 


*In our world the deepest knowledge is not in words. It is in the meaning behind the words, in the spaces between them, in gestures or looks, in*
*meaningful silences, in the work of hands, in learning from journeys, in quiet reflection, in the Dreaming.*

*([67], p. 39)*


Thus, the concept also embodies the sense of critical thinking as a person fosters their observational and analytical skills, without reliance on open questioning (if you need to ask a question, it may show that you haven’t listened properly or are relying on someone else to do the thinking’), and is a highly valued capacity within the Yolŋu worldview. *Nhina, nhäma ga ŋäma* can also be applied as a form of reflexivity, as one reflects on, listens to, and observes the self.

### 3.4. Alignment with Non-Aboriginal Research Methods

*Nhina, nhäma ga ŋäma* can be broadly applied as both methodology and method. It can be understood as closely aligned to the Western academic concept of participant observation, it can be a stance adopted in the yarning process to facilitate empathy and listening, and it can be applied to data analysis as participants’ emotions and meanings are carefully attended to. Combined with yarning it contributes to a rigorous multi-layered ethnographic methodology. Even to learn about *nhina, nhäma ga ŋäma* requires taking an experiential and reflective approach, described by MM as ‘close your eyes and the answer will come to you’ (Researcher journal note, October 2016).

In essence, *nhina, nhäma ga ŋäma* as a research method goes beyond listening/understanding the speech of the person with whom you are communicating, to being aware of their true (deep) feelings. ‘Putting interest in other person, do it with respect, you can feel while you listen, you can feel something. The feeling tells you when you listen’ (Yolŋu co-researcher). It also requires the researcher to have self-awareness: ‘If you have no feelings then you can’t tell the feelings of the other person. Or you can’t feel other people when they’re telling stories’ (Yolŋu co-researcher). Self-awareness and reflexivity are also required when listening with an open heart. Thus, researchers operate in a culturally appropriate manner by asking few/no questions, instead observing and reflecting on the possible feelings embodied in the actions and conversations being observed, thus building empathy/rapport. 

The Yolŋu co-researchers were not familiar with the term empathy, but through the research it became clear this is conceptually closely aligned to the role of *nhina, nhäma ga ŋäma* in terms of understanding feelings. In this way, the Yolŋu identified that research rigour, a sense of having strong generalisable evidence, is based on stories that ‘come out from the heart as well as the mind’ (Researcher journal note, 7/4/17). That is, one will know when one has the ‘true story’ by how it feels, a notion explored with Dr Lawurrpa Maypilyama, a senior Yolŋu woman and experienced researcher from Galiwin’ku. She described it thus: 


*‘People want to share their stories, the deep meaning … a true story has to be strong, the listener can feel when it’s the truth’. ‘Yolŋu tell a story not by head but by heart–if I tell you from the head you won’t feel anything. But if I have excitement, expression, you can tell it is a true story.*

*(Researcher journal note 16/5/17)*


This suggestion is a potential source of tension between an emphasis in Western qualitative research on interview data that prioritises the authority of words as evidence. Whereas for the Yolŋu true knowledge is about feelings as a result of listening and watching *‘and it is about sacrifice ... You have to put yourself aside to really sit quietly and listen, and then knowledge will come to you’* (YG, Researcher journal note, 11/8/17). 

### 3.5. Metaphors 

The important place of metaphor in Aboriginal knowledge systems is widely recognised [20,68], particularly as a conceptual meaning-making tool. Metaphors are central to storytelling; using metaphor has the potential to create both a feeling (i.e., it conveys emotion) and a common ground. ‘Yolŋu have used metaphors since time immemorial to share strong messages about how we should live and work ... for “understanding and experiencing one kind of thing in terms of another”’ ([69], p. 406). 

Concern that a non-Aboriginal lens might reduce metaphors to figures of speech has led others to suggest using terms such as ‘images’ or ‘symbols’, as these ‘must be experienced in order to be understood, and the experience of its effects is at once its meaning and its power’ ([70], p. 216; [71]). Understanding metaphor-as-symbol connects to a broader discussion regarding the use of metaphors for learning, research and knowledge acquisition across all languages and cultures. The Yolŋu process of ‘reading the signs’, that is observing and correctly interpreting ‘indicators’ such as the presence of a particular flower (indicating to the Yolŋu that it is the time for hunting for certain fish), demonstrates a parallel to Western biomedical scientific procedures such as understanding blood glucose levels (indicating risk of disease). 

The privileging of mainstream science has at times silenced Aboriginal ontologies [69,70], and in the article co-authored by three senior Yolŋu women, they argue that to counter this requires ‘taking [Yolŋu] metaphors seriously’ as a means to effectively privilege Aboriginal knowledge, values, attitudes, and practices [69]. As a means of broadening the space for productive dialogue, metaphors work towards ‘making space for difference’ [69]. That is, ‘metaphorical language is theoretical. It stimulates analytical and creative possibilities’ ([71], p. 14), where ‘one will eventually see more than what is presented’ [72], p. 229. The use of metaphor as an effective means of communication in clinical conversations [73], education [74], and in discussing health systems [75] is also recognised.

### 3.6. Applying Metaphors in the RHD Research Context

Developing concept bridges through metaphors is a central both-way learning tool. Critical thinking and productive dialogue are essential to developing the metaphors, symbols, and images that build conceptual links across cultures, and is a central theme throughout the research reported here. 

Two key metaphors are described here. One from the start and the other from the very end of the research project. These metaphors were built up over many productive, iterative discussions (the extent to which a metaphor can be extended is a marker of the strength and value of the metaphor). Permission was gained from community elders for the use of the cultural knowledge contained in these. While these examples provide evidence of a process, the content should not be assumed to be transferable to other settings.

The first important and ongoing metaphor was developed at the start of a community-based action research project where the skills developed through the project were aligned with a Certificate II in Community Health Research [50]. At the beginning the community was not familiar with the term ‘research’. The remote communities of north-east Arnhem Land have limited experience with outside researchers. The knowledge needed to guide a *lipa lipa* (canoe) was found to be a good both-way learning bridge to align Yolŋu ways of observing and analysing data (signs) from their environment with Western research terms. In doing so we came to define research conceptually as looking and thinking deeply, as in guiding the direction of the *lipa lipa*, the leader (researcher) must look at more than just the surface of the water, looking up to consider factors such as wind direction and below the surface to identify currents and hidden dangers. Central to the development of this metaphor were many long conversations about the link between research as critical thinking and asking the right questions, with a definite sense from the Yolŋu that the research question should be practical and change-oriented. Critical thinking and asking the right question about how things might be different is inherently transformative, demonstrating the empowering effects of developing research skills. 

Later a cohort of community members doing the Certificate II developed further research skills through their involvement in the lived experience of RHD PhD research project. In the course of analysing the yarns, a weaving metaphor for research was developed, initially for a both-way understanding of what thematic analysis and coding meant. This important metaphor stood the test of being retold and added to many times, including with other more experienced Yolŋu researchers from Galiwinku. Table 1 below describes the full metaphor. 

## 4. Discussion

We argue that it is naïve to think that applying First Nations knowledge practices ‘necessarily divorces the researcher from the colonised and hegemonic space of Western research philosophies’ ([76], p. 45). First Nations knowledge practices cannot simply be slotted into or alongside mainstream research practices to justify a research process. Similarly, trying to replace mainstream with First Nations epistemologies would likely perpetuate the simplistic, oppositional binary thinking of colonialism and the Western academy ([77], p. 104). Instead, as we have demonstrated here, it is possible to have a process that acknowledges and merges different knowledge systems through well-articulated and respectful consideration that keeps First Nations ways of knowing central, but also aligns with mainstream epistemologies both in the structure of the research and in the research procedures [55].

The learnings reported here from the Yolŋu regarding relationality, *nhina, nhäma ga ŋäma*, and metaphors as basis for critical/conceptual thinking, demonstrate that First Nations and mainstream research methodologies are not mutually exclusive and can be used effectively to complement and support each other. There are, however, challenges and tensions in this.

Each researcher in the project described here needed to hold the stance of *nhina, nhäma ga ŋäma*, enabling dialogue across two very different linguistic and cultural backgrounds and life worlds. Dialogues included an equal valuing of everyone’s contribution, paying attention to relationships, accepting philosophical discomfort, and thinking critically about power dynamics, all contributing to writing with a transformative purpose. Central to dialogue is attention to difference; it is only when the productive potential of difference is emphasised that the actual work of collaboration is achieved. In terms of approaches to research, the Yolŋu have insider knowledge that is personal, familial, and cultural. This goes beyond empathy. It reflects that the cohering idea of an Aboriginal worldview is the interconnection of everything, therefore Aboriginal research is always holistic. Even ideas are taken to be animate [78] and interconnected with the world, ‘thus, the Indigenous scholar has to consider and represent how they are part of that idea, rather than how they can order and manipulate it’ [79]. As described in the Results, this was exemplified by the rapidity with which Yolŋu co-researchers were able to arrive at themes. 

There were many other varied differences that prompted reflection and dialogue–language (including speaking many languages), education type, experiences of Australian government policies and colonisation, beliefs regarding health and wellbeing, degree of connection to country and family, exposure to environmental health risks, and capacity to perceive the world holistically. In addition, beyond the differences within the group of Aboriginal and non-Aboriginal social science researchers, we were aware of the need to bridge the divide between the lived experiences of the Yolŋu with RHD and the biomedical worldview. Enabling clinicians to regard Yolŋu ways of knowing being and doing with equal value, despite its unfamiliarity, and to build their confidence in working in a new way was our hope for better provision of healthcare for Aboriginal people. This requires an appreciation of a worldview that prioritises relationships, empathy, critical thinking, and the well-being of others (family, community, and land). 

Not previously heard in the RHD domain, the findings from the research based in the methods described here reveal fundamental differences between Aboriginal and biomedical worldviews contributing to the failure of current approaches to communicating health messages. The themes identified by the Yolŋu co-researchers (for details see [53]) affirmed an Aboriginal worldview and demonstrate the inter-connectedness of knowledge, choice, and behaviour that become increasingly complex in stressful and traumatic health, socioeconomic, political, historical, and cultural contexts. In response, the Yolŋu also provided targeted recommendations for culturally-responsive health promotion, including: communicating to create positive emotions; building trust; and developing health messages (including relevant data) that are aimed at the family and community rather than at individuals.

As described in the last stage of the weaving metaphor (Table 1), the Yolŋu were highly motivated by sharing knowledge about what action they want, ‘so everyone can benefit’, to use the findings from research to create ‘a mat for all the family to sit on’. In this discussion, they recognised that it was necessary to make a different version of results for different people (community members, scientists, health workers, and so on). It was encouraging, therefore, to see the powerful and pragmatic lived experience findings and recommendations used in a wide variety of knowledge translation modes including: small community initiatives (a baby’s healthy skin book), academic papers, online newspapers [80], a thesis [64], conference presentations, lectures to medical and public health students, integration into the RHD Clinical Guidelines [81], and the Endgame Report [82]. Further broader initiatives include supporting the RHDA Champions4change program (https://www.rhdaustralia.org.au/champions4change, accessed on 8 March 2022) and new projects that provide opportunities for building capacity with First Nations co-researchers.

These practical outcomes are encouraging as they reflect that First Nations and Western research methodologies were effectively combined to complement and support each other through a process of respectfully acknowledging and merging different knowledge systems. However, on their own, these are small steps; radical funding and policy shifts are needed to upscale and embed approaches that prioritise place-based First Nations leadership and governance, funding for primordial prevention, community development, and environmental health on a par with biomedical research such as that for vaccine development.

First Nations Community Research Capacity Building 

To allow for future projects we propose a both-way learning model (Figure 2) in order to generalise from and apply the learnings regarding building community research capacity. This model is designed to be applied in small-scale, place and strength-based, community-led actions that can then be co-evaluated using critical, decolonising social science approaches, guided by First Nations knowledges and worldviews. Once initiated, and after community confidence in newly acquired knowledge has grown, it is likely that local communities will initiate actions to disseminate learnings and messages more broadly [50].

## 5. Conclusions

Through the research collaborations described above, co-researchers had opportunities to disrupt colonial narratives and create new possibilities to transform, policies, practice, and systems [83]. By broadening the space of productive dialogue, a mat we can all sit on, new positions, views, and identities emerged. Through productive dialogue, cultural differences were moved from being a ‘problem’ or incommensurable to a more powerful discourse that allowed for sharing of ideas and robust and highly respectful discussions that engaged with rather than resisted cultural differences, offering hope as well as new insights and new opportunities. In particular, we hope the results motivate clinicians to regard Yolŋu ways of knowing being and doing with equal value, despite its unfamiliarity, and to build their confidence in working in new ways for better provision of healthcare for Australian Aboriginal people.

This paper has described how a suitable methodology was developed to undertake meaningful research into the lived experience of RHD. Our companion paper [52] describes how we used this approach and the detailed findings emerging from the data collected. 

## Figures and Tables

**Figure 1 ijerph-19-03654-f001:**
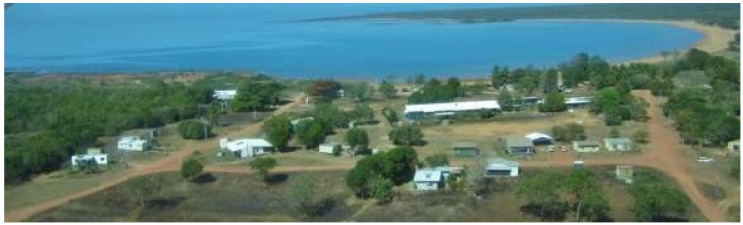
Yilpara viewed from the air (*photo by author*).

**Figure 2 ijerph-19-03654-f002:**
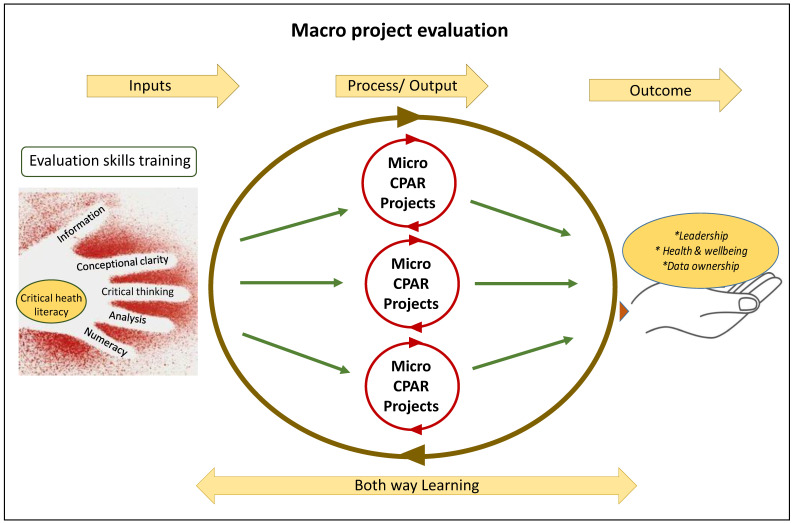
Both-Way learning model.

**Table 1 ijerph-19-03654-t001:** Weaving and qualitative research metaphor bridge.

Weaving	Qualitative Research
Go out looking for pandanus	Go out to look for stories
Find good (right one) pandanus	Choose who to interview, do it the right way (consent)
Collect pandanus—choose long central leaves, carry a special stick to get the tall pandanus, collect pandanus that is right for the item that is being made	Choose the right research tools to use and use them correctly, for example, take a translator if doing in-depth interviews
Sort the pandanus (feeling with hands, looking with eyes)–take off the spikes, split/peel the leaf Keep some, throw some out discard others	Coding—decide what to keep, what to throw out discard or put aside
Choose which pandanus leaves to colour Prepare the colour Find some ash to add to make the colour stronger	Colour code Or drag and drop into nodes (NVivo) (add theory/other research to make the codes (colour) stronger)
Colour the pandanus	Themes drawn from codes
Weave the pandanus Make patterns with the different colours Make a basket or mat	Weave the themes into a clear story
Take your basket to show people your work Make a mat that the whole family can sit on Make something of use to everyone Dilly bag for ceremony is full of story in two moieties Fish nets are another example	Sharing knowledge about what action we want with everyone, so everyone can benefit (knowledge translation) *‘a mat for all the family to sit on’* Make a different version of results for different people (community, scientist, health workers) Combine quantitative and qualitative data

## Data Availability

Thesis contains more detailed data, full de –identified interviews available on request.

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
