# Peer review of "“Weaving a Mat That We Can All Sit On”: Qualitative Research Approaches for Productive Dialogue in the Intercultural Space"

_ijerph, 2022, doi:10.3390/ijerph19063654_

Round 1

Reviewer 1 Report

The authors reported lessons learned from a case study focusing on rheumatic heart disease among Aboriginal Australians. These lessons are very important and the authors proposed important lessons that can be adopted when conducting a qualitative study among the Aboriginal people which could potentially “decolonize” the research process and balance the power dynamics between the participants and the researchers.

Overall, the whole manuscript is very well-written and provided an important insight in using metaphors which would promote dialogue between the First Nations community and the researchers leading to a more meaningful and inclusive research findings.

I have no major comments this very well-written article except for final proof-reading for minor typos throughout the whole manuscript. For example, the first word in line 613 was not capitalized.

Author Response

Thank you, we have completed a detailed proof read. 

With regard to your comment that "the first word in line 613 was not capitalized", in fact the start of the sentence was cut off. The following text has now been added,

“Through the research collaborations described above, co-researchers had opportunities to ....”

Reviewer 2 Report

This is a very interesting, important and timely article on how to do collaborative (biomedical) research and decolonise Western research methodologies. It is well-written and provides a clear and engaging overview of the specific research methods that were employed to enable a better understanding of the lived experience of rheumatic heart disease, which is a marker of health inequity for First Nations Australians. The article outlines the requirement of long-term engagement and building of trust (including reciprocity, relationality, empathy) between non-indigenous and first nations peoples before research can take place, and the importance of local vernacular and methaphors in order to successfully 'co-mingle' western research design and knowledge with local ontologies and understandings.  

A few minor edits:

p. 6 line 233, add full stop after [34]

p.7 line 274, add space after 'each other'

p.8 line 371, This rapidity also contradicts presumptions of research funding and donors that collaborative research takes too much time.

 p.12 line 514-524, I think the article can do without the table of the Kakala framework as it distracts from the Yolnu case and poses the question as to why there is reference or comparison with this particular model only and no engagement with other examples? Pacific scholars have been on the forefront of decolonising western dominated research and academia. I would mention a few other examples and remove the Kakala table. There are also other examples, including non-medical research collaborations in Aboriginal communities. See for example: MEDIATING ACROSS DIFFERENCE: Oceanic and Asian Approaches to Conflict Resolution. Writing Past Colonialism Series. Edited by Morgan Brigg and Roland Bleiker, which might be helpful.

p.14 line 613. Start of first sentence is missing

Reviewer 3 Report

It was a pleasure reading this excellent paper, rich in cross-cultural ideas and especially in methodological approaches and epistemic perspectives. Clever and creative at the same time. I dont have the chances to review too many articles like this. As a sociologist also interested in developing new methodologies (in particular related to the power of personal narratives) and in decolonising research methodologies and (death studies) theories I had so many things to learn. Your article is thought- provoking at some many levels and it made visible for me how much power lie in the instruments through which we understand/meseaure our social world. I will surely look for the subsequent article of yours where the research results regarding the lived experiences of the rheumatic heart disease persons are presented. Just one minor issue: please go throught the article because there seem to be many extra blank spaces between the words. 

Author Response

Thank you for your appreciative review.   We have also had only minor revisions on the second article reporting the research results regarding the lived experiences of  rheumatic heart disease, so that will hopefully be published soon in the same edition of the journal as the one you reviewed. 

In response to your comment - I have gone through the article to check for white spaces (I think they came in when the article was converted by the journal).  Thank you